# GAUSSIANFOCUS: CONSTRAINED ATTENTION FOCUS FOR 3D GAUSSIAN SPLATTING

## ABSTRACT

Recent developments in 3D reconstruction and neural rendering have significantly propelled the capabilities of photo-realistic 3D scene rendering across various academic and industrial fields. The 3D Gaussian Splatting technique, alongside its derivatives, integrates the advantages of primitive-based and volumetric representations to deliver top-tier rendering quality and efficiency. Despite these advancements, the method tends to generate excessive redundant noisy Gaussians overfitted to every training view, which degrades the rendering quality. Additionally, while 3D Gaussian Splatting excels in small-scale and object-centric scenes, its application to larger scenes is hindered by constraints such as limited video memory, excessive optimization duration, and variable appearance across views. To address these challenges, we introduce GaussianFocus, an innovative approach that incorporates a patch attention algorithm to refine rendering quality and implements a Gaussian constraints strategy to minimize redundancy. Moreover, we propose a subdivision reconstruction strategy for large-scale scenes, dividing them into smaller mergeable blocks for individual training. Our results indicate that GaussianFocus significantly reduces unnecessary Gaussians and enhances rendering quality, surpassing existing State-of-The-Art (SoTA) methods. Furthermore, we demonstrate the capability of our approach to effectively manage and render large scenes, such as urban environments, maintaining high fidelity in the visual output. (The link to the code will be made available after publication)

## 1 INTRODUCTION

Novel View Synthesis (NVS) is fundamental for modern computer graphics and vision, extending to virtual reality, autonomous driving, and robotics. Primitive-based models such as meshes and point clouds (Lassner & Zollhofer, 2021; Munkberg et al., 2022; Yifan et al., 2019), optimized for GPU rasterization, deliver fast but often lower-quality images with discontinuities. The introduction of Neural Radiance Fields (NeRF) by (Mildenhall et al., 2021) marked a significant advancement, employing a multi-layer perceptron (MLP) to achieve high-quality, geometrically consistent renderings of new viewpoints. However, NeRF's reliance on time-consuming stochastic sampling can lead to slower performance and potential noise issues.

Recent advancements in 3D Gaussian Splatting (3DGS) (Kerbl et al., 2023) have significantly enhanced rendering quality and speed. This technique refines a series of 3D Gaussians initialised with using Structure from Motion (SfM) (Snavely et al., 2006) to model scenes with inherent volumetric continuity, facilitating fast rasterization by projecting onto 2D planes. However, 3DGS often produces artifacts when camera viewpoints deviate from the training set and lack detail during zooming. To address these issues, newer models (Yu et al., 2024; Lu et al., 2024) employ a 3D smoothing filter to regularize the maximum frequency and utilize anchor points to initialize 3D Gaussians, thereby enhancing visual accuracy and applicability in diverse scenarios. Despite these advances, 3DGS-based models still tend to use oversized Gaussian spheres that ignore scene structure, leading to redundancy and scalability issues in complex environments. Additionally, these models struggle with detail reconstruction, particularly at edges and high-frequency areas. This often leads to suboptimal rendering quality. Moreover, reconstructing large-scale scenes like towns or cities represents a significant challenge due to GPU memory constraints and computational demands. To mitigate these problems, models often reduce training input randomly, which compromises reconstruction quality and results in incomplete outcomes.

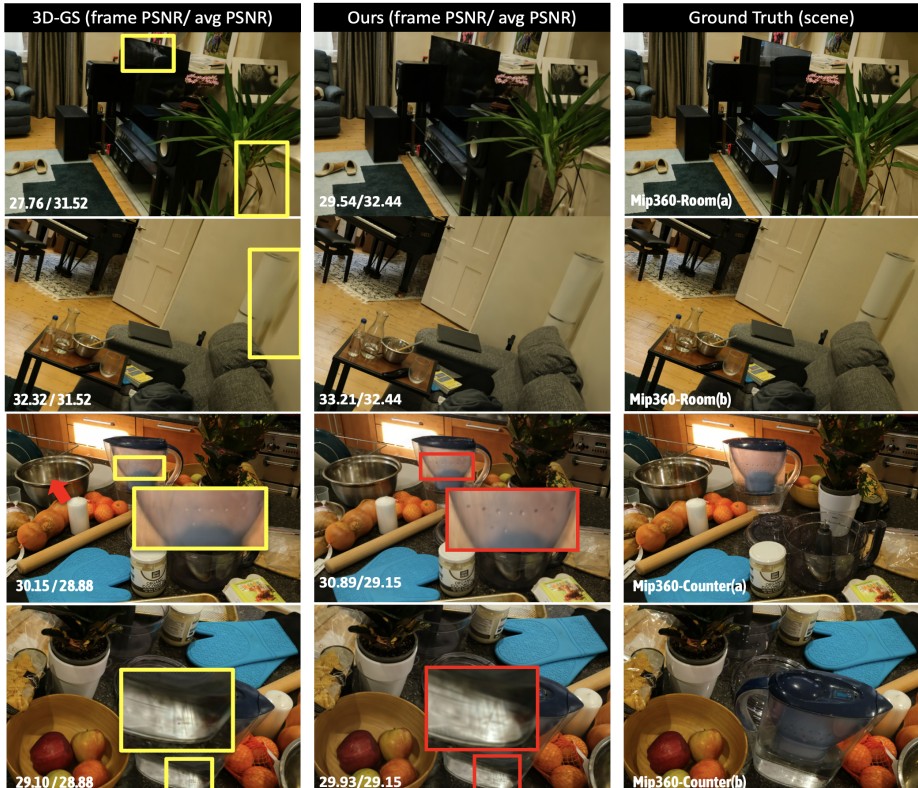

Figure 1: **GaussianFocus.** As illustrated by the red and yellow boxes in the images, our method consistently surpasses the 3DGS model in various scenes, showing distinct advantages in challenging environments characterized by slender geometries, intricate details, and lighting effects.

To address quality issues in 3D Gaussian Splatting (3DGS), we introduce GaussianFocus, a framework designed for enhanced fidelity in both general and large-scale scene reconstructions. GaussianFocus employs a patch attention algorithm and Sobel operators to refine edge details and spatial frequency during training, thereby improving scene fidelity. We also apply constraints on the size of Gaussian spheres during initialization and training phases, which refines texture details and diminishes the occurrence of "air walls". These "air walls" are spurious barriers or noise in 3D reconstructions, typically resulting from oversized Gaussian spheres that disrupt visual coherence. For reconstructing extensive scenes, our method uses bounding boxes to divide each scene along the XYZ axes into manageable blocks. Each block is independently processed in our 3D reconstruction pipeline, ensuring precise attention to its specific features. After processing, these blocks are seamlessly recombined, producing a coherent and detailed large-scale reconstruction.

Through rigorous experiments, our GaussianFocus model has outperformed traditional 3DGS models (Kerbl et al., 2023), as evidenced in Fig. 1. It notably reduces artifacts associated with oversized Gaussian spheres, thereby enhancing the quality of 3D reconstructions. Our subdivision strategy for large-scale scenes considerably lowers GPU computational demands, allowing for the use of all input data and maintaining superior reconstruction quality. This represents a significant improvement over previous approaches (Kerbl et al., 2023; Yu et al., 2024; Lu et al., 2024; Guédon & Lepetit, 2024), which often required sub-sampling of input data to manage computational loads. GaussianFocus thus significantly improves the realism and quality of 3D reconstructions.

In summary, the contributions of this work are as follows:

1. We propose a 3DGS-based patch attention algorithm with designed edge and frequency losses to enhance the details and reduce spatial frequency artifacts within scene reconstructions. This improves the detailing quality and intricacy of the rendered scenes.

2. We impose constraints on overly large Gaussian spheres to mitigate the occurrence of "air walls", thus refining the scene reconstruction's fidelity and enhancing the granularity of the resulting models. Moreover, these constraints allow the achievement of superior reconstruction results with fewer training iterations.

3. For large-scale scene reconstruction, our approach involves subdividing the scene for individual reconstruction and subsequent recombination. This method addresses the challenge posed by existing 3DGS-based models that fail to directly reconstruct extensive scenes, thereby enhancing the scalability and applicability of our reconstruction framework.

In this paper, we structure the content as follows: Section 2 indicates the preliminary concepts. Section 3 outlines the methods we employed. In Section 4, we present our experimental framework compare its performance to other advanced 3DGS-based models and discuss the ablation studies. We conclude the paper in Section 5. For a review of related work of our paper, implementation details and model limitation, please refer to Appendix A.

## 2 PRELIMINARIES

In the foundational aspects of the 3DGS framework (Kerbl et al., 2023), the scene is represented using anisotropic 3D Gaussians that integrate differential properties typical of a volume-based approach but are rendered more effectively through a grid-based rasterization technique. Beginning with a collection of structure-from-motion (SfM) (Snavely et al., 2006) data points, each point is established as the centroid ($\mu$) for a 3D Gaussian. The formula for a 3D Gaussian $G(x)$ is given by:

$$G(x) = \exp\left(-\frac{1}{2}(x - \mu)^T \Sigma^{-1}(x - \mu)\right), \tag{1}$$

where $x$ represents a point within the 3D space, and $\Sigma$ represents the Gaussian covariance matrix which is constructed using

$$\Sigma = RSS^T R^T. \tag{2}$$

This configuration is derived from a scaling matrix $S$ and a rotational matrix $R$, guaranteeing its positivity and semi-definiteness.

Each Gaussian is not only linked with a colour $c_i$, defined through spherical harmonics but also paired with an opacity $\alpha$, impacting the merging process in rendering. Diverging from classic volumetric methods that employ ray-marching, this model projects 3D Gaussians onto a 2D plane $G^{2D}(x)$ and processes them through a grid-based rasterizer for sorting and $\alpha$-blending. The $\alpha$-blending formula is specified as

$$C(x') = \sum_{i \in K} c_i \sigma_i \prod_{j=1}^{i-1}(1 - \sigma_j), \tag{3}$$

where $\sigma_i = \alpha_i G_i^{2D}(x')$, $x'$ represents the specified pixel position, and $K$ counts the Gaussians for that specified pixel in two dimensions. This approach facilitates the direct learning and optimization of the Gaussian features through a trainable differentiable rasterizer.

## 3 METHODOLOGY

The traditional 3DGS (Kerbl et al., 2023) and its variants (Yu et al., 2024; Guédon & Lepetit, 2024; Lu et al., 2024) employ Gaussian optimization to reconstruct scenes, often failing to accurately represent actual scene structures and struggling with oversized Gaussians that blur scenes and lead to information loss. Limited GPU memory and extended optimization times further hinder their ability to reconstruct large scenes. Our enhanced framework, detailed in Fig. 2, addresses these limitations by imposing constraints on the size and quantity of 3D Gaussian spheres, reducing redundancy and improving robustness against varying viewing conditions. We incorporate attention mechanisms and a combination of edge and frequency loss to refine reconstruction quality.

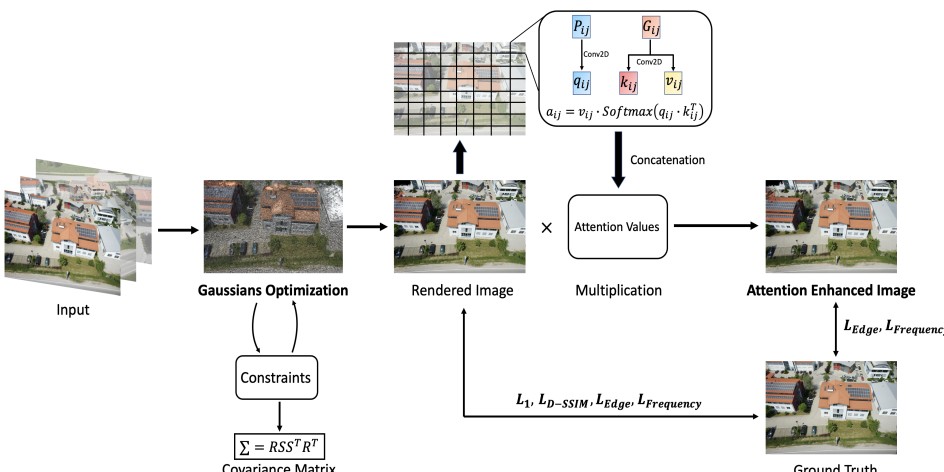

Figure 2: **Overview of GaussianFocus:** Our model will monitor the size of Gaussian spheres during initialization and training. **Constraints** are applied to the scaling matrix $S$ within the covariance matrix to prevent Gaussian spheres' excessive growth. Subsequently, the rendered image is divided into 64 parts. Each part independently calculates its attention values, which are then concatenated to form a comprehensive attention map. This map is multiplied back onto the original rendered image to produce an **attention-enhanced image**. Finally, this enhanced image and the original rendered image undergo multiple loss calculations against the ground truth. These include reconstruction ($L_1$), structural similarity ($L_{D-SSIM}$), edge ($L_{Edge}$), and frequency ($L_{Frequency}$) losses.

## 3.1 3D GAUSSIAN-BASED PATCH ATTENTION ENHANCEMENT

Given the significant computational demands, it is impractical to directly compute attention values for the entire rendered image due to the extensive data processing involved. Instead, both model-rendered image $P_i$ and the Ground Truth images $G_i$ are segmented into 8x8 regions to manage computational complexity effectively. For each segment of $P_i$, a query vector $q_{ij}$ is extracted using a 2D convolutional layer which is designed to capture detailed features and spatial relationships within the segment. Correspondingly, the key $k_{ij}$ and value $v_{ij}$ for each segment $j$ of $G_i$ are derived through similar 2D convolutional layers. These steps ensure that the essential components for the multi-head attention mechanism—queries, keys, and values (QKV)—are accurately assembled based on localized image features. The attention weights $w_{ij}$ for each segment can be calculated using the following equation:

$$w_{ij} = \text{Softmax}(\alpha_{ij}), \quad \alpha_{ij} = q_{ij} \cdot k_{ij}^T, \tag{4}$$

where $\alpha_{ij}$ represents the unnormalized attention scores, which are computed as the dot product of the query and the transposed key. This product measures the compatibility between different parts of the image, facilitating a focused synthesis of features. The attention map for each segment $a_{ij}$ is generated by applying the weighted sum of the values using the attention weights:

$$a_{ij} = v_{ij} \cdot w_{ij}, \tag{5}$$

where $w_{ij}$ scales the value $v_{ij}$ according to the relevance of each segment's features, thereby producing a segment-specific attention map that highlights pertinent features. Concatenating these individual attention maps yields a comprehensive attention map $A_i$ for the image, which can be represented by:

$$A_i = \bigoplus_j a_{ij} \tag{6}$$

where the sum over $j$ aggregates the contributions of all segments into a unified attention profile for the entire image. This comprehensive attention map $A_i$ is then used to produce an attention-enhanced image $P_i^{'}$ by element-wise multiplying it with the rendered image $P_i$:

$$P_i^{'} = P_i \otimes A_i, \tag{7}$$

Figure 3: **Subdivision-Based Reconstruction of Large Scenes Procedure.** Our method divides large scenes into blocks for reconstruction.

which enhances the original image by amplifying features that are deemed significant based on the attention mechanism. To further enhance the reconstruction's accuracy, we compute edge loss $L_{Edge}$ and frequency loss $L_{Frequency}$ for this enhanced image in conjunction with the ground truth image. These losses are calculated alongside the standard loss comparisons between the original rendered image and the ground truth image. They will be discussed in Section 3.4.

## 3.2 GAUSSIAN SPHERE CONSTRAINTS

During the initialization of Gaussian spheres, we impose constraints on the scaling matrix $S$ to control the covariance matrix's influence, essential for accurately modelling spatial relationships in the scene. The adjustment of $S$ is dictated by the density of the initial point cloud data: for denser point clouds, we set a lower initial scaling value to reduce overlaps and redundancy, while for sparser distributions, we increase it to ensure sufficient scene coverage. This careful calibration of scaling factors helps maintain an optimal balance between preserving detail and enhancing computational efficiency. The scaling matrix constraint is defined as follows:

$$S_i = S_i \cdot \alpha, \quad \text{if } S_i > \tau, \tag{8}$$

where $S_i$ denotes the scales in the scaling matrix of the Gaussians. The $\tau$ serves as a threshold scale and $\alpha$ is a modulating factor, both of them adjusted experimentally. In our experiment, we set $\tau = 0.3$ and $\alpha = 0.2$. The adaptive scaling in our model not only mitigates computational load but also aligns with the varying densities of real-world data. Enhancing the traditional "split and clone" strategy of the 3DGS (Kerbl et al., 2023) model, we integrate a filtering mechanism to manage excessively large Gaussians during training. This involves implementing a selection criterion to identify large Gaussians post-splitting, followed by a strategic reduction in their scale. Additionally, we employ a selective splitting strategy for older Gaussians that have remained in the model over extended periods. This technique is based on both the age and the operational efficiency of the Gaussian in terms of scene representation:

$$\text{Selective Split } (S_\gamma), \quad \text{if } S_\gamma > \Omega \tag{9}$$

where $S_\gamma$ denotes the scales of the scaling matrix of aged Gaussians and $\Omega$ is the threshold set to identify old Gaussians that require reevaluation. We set $\Omega = 0.3$ in our experiment. These strategies ensure that our method maintains a balanced approach to managing the size and number of Gaussians within the 3DGS framework.

## 3.3 SUBDIVISION-BASED RECONSTRUCTION OF LARGE SCENES

In response to 3DGS challenges (Kerbl et al., 2023; Yu et al., 2024; Lu et al., 2024), our method initiates a preprocessing step to acquire initial points from Structure-from-Motion (SfM) (Snavely et al., 2006) of the large scene. As shown in Fig. 3, a three-dimensional bounding box is then constructed to encompass all initial point clouds. We divide this bounding box along its $xyz$ axes into $n \times n \times n$ blocks, where each block is defined to contain its respective subset of point clouds:

$$B_{ijk} = \{pc \in \text{Point Cloud} : (x_i \leq pc_x < x_{i+1}) \land (y_j \leq pc_y < y_{j+1}) \land (z_k \leq pc_z < z_{k+1})\}, \tag{10}$$

where $pc$ represents a point in the point cloud and $x_i, y_j, z_k$ denote the boundaries of block $B_{ijk}$. We have integrated a distance iteration algorithm to address the potential for sparse outlier points

| | SSIM ↑ | | | | | PSNR ↑ | | | | | LPIPS ↓ | | | | |
|---|---|---|---|---|---|---|---|---|---|---|---|---|---|---|---|
| | Original Res. | 1/2 Res. | 1/4 Res. | 1/8 Res. | Avg. | Original Res. | 1/2 Res. | 1/4 Res. | 1/8 Res. | Avg. | Original Res. | 1/2 Res. | 1/4 Res. | 1/8 Res. | Avg. |
| NeRF | 0.933 | 0.966 | 0.970 | 0.948 | 0.954 | 31.27 | 31.98 | 29.98 | 26.52 | 29.94 | 0.059 | 0.040 | 0.049 | 0.059 | 0.052 |
| Mip-NeRF | 0.960 | 0.968 | 0.970 | 0.960 | 0.965 | 32.50 | 33.00 | 31.20 | 28.10 | 31.20 | 0.044 | 0.030 | 0.035 | 0.051 | 0.040 |
| Instant-NGP | 0.963 | 0.968 | 0.965 | 0.946 | 0.961 | 33.05 | 33.10 | 29.80 | 26.45 | 30.60 | 0.046 | 0.036 | 0.048 | 0.072 | 0.051 |
| TensoRF | 0.958 | 0.970 | 0.960 | 0.950 | 0.960 | 32.60 | 32.75 | 30.20 | 26.30 | 30.46 | 0.046 | 0.035 | 0.047 | 0.070 | 0.050 |
| Tri-MipRF | 0.961 | 0.969 | 0.953 | 0.908 | 0.948 | 32.75 | 33.00 | 29.70 | 24.10 | 29.89 | 0.048 | 0.038 | 0.048 | 0.072 | 0.051 |
| 3DGS | **0.973** | 0.952 | 0.868 | 0.761 | 0.889 | 33.50 | 27.10 | 21.60 | 17.80 | 25.00 | 0.032 | 0.022 | 0.068 | 0.118 | 0.060 |
| 3DGS + EWA | 0.967 | 0.974 | 0.955 | 0.943 | 0.960 | **33.60** | 31.80 | 27.95 | 24.75 | 29.53 | 0.035 | 0.026 | 0.036 | 0.049 | 0.037 |
| Ours | 0.971 | **0.975** | **0.972** | **0.975** | **0.973** | 33.29 | **33.96** | **31.64** | **28.65** | **31.89** | **0.031** | **0.017** | **0.023** | **0.028** | **0.025** |

Table 1: **Quantitative Comparison with Baselines on the Blender Dataset (Mildenhall et al., 2021).** All models are evaluated at four progressively lower resolutions and trained using images at original resolutions. Our method outperforms other models at 1/2, 1/4, and 1/8 resolutions and achieves comparative results at the original resolution.

to skew the subdivision logic. This algorithm iterates through all points, identifying and discarding those that do not contribute meaningfully to the division process:

$$\text{Iterate } \forall pc \in \text{Point Cloud} : \text{if } \text{dist}(pc, \text{Block}_{ijk}) > \theta \text{ then discard } pc, \tag{11}$$

where $\text{dist}(\cdot)$ calculates the distance from the point to the nearest block boundary, and $\theta$ is a threshold value defining the maximum allowable distance for inclusion. Corresponding camera and Structure-from-Motion (SfM) points associated with each block are classified to assemble the essential initial files required for training. Each block undergoes independent training. The process concludes with the recombination of the divided scene's 3D files, thus completing the reconstruction of the entire large scene. This modular approach alleviates the computational and memory constraints typically linked with large-scale scene reconstruction. By employing this method, we efficiently manage large scene datasets and enhance the scalability of our reconstruction processes.

## 3.4 TRAINING LOSSES

In our GaussianFocus model, following 3DGS, the loss function incorporates both L1 and D-SSIM terms. The L1 term measures absolute differences between predictions and targets, while D-SSIM enhances perceptual image and video quality. To improve the structural accuracy during training, we designed an edge loss term that leverages the Sobel operator to extract edge information effectively. This operator is applied to each channel of both the input and target images to compute their respective gradients in the $x$ and $y$ directions. The edge loss is then calculated as the average of the L1 loss of these gradients:

$$L_{\text{Edge}} = \frac{1}{2} \left( \text{L1}(\nabla_x p_i, \nabla_x \hat{p}_i) + \text{L1}(\nabla_y p_i, \nabla_y \hat{p}_i) \right), \tag{12}$$

where $\nabla_x$ and $\nabla_y$ represent the gradient operator calculated using the Sobel filter, capturing edge information along the $x$ and $y$ directions. The $p_i$ and $\hat{p}_i$ represent the pixels of the ground truth image $G_i$ and the corresponding pixel in the rendered image $P_i$, Moreover, we introduce the frequency loss term to address the challenge of high-frequency detail loss. It approximates the frequency domain loss by employing gradient loss computations in the $x$ and $y$ directions for both the input and target images. This term is essential for preserving high-frequency details and is computed as:

$$L_{\text{Frequency}} = \frac{1}{2} \left( \text{L1}(G_x(p_i), G_x(\hat{p}_i)) + \text{L1}(G_y(p_i), G_y(\hat{p}_i)) \right), \tag{13}$$

where $G_x$ and $G_y$ are the changes in pixel values along the horizontal and vertical axes. The overall loss function for the GaussianFocus model integrates these individual loss components into a weighted sum, optimizing the reconstruction quality across multiple dimensions:

$$L_{\text{Total}} = \begin{cases} (1-\lambda)L_1(p_i, \hat{p}_i) + \lambda L_{\text{D-SSIM}}(p_i, \hat{p}_i) + \beta L_{\text{Edge}}(p_i, \hat{p}_i') + \eta L_{\text{Frequency}}(p_i, \hat{p}_i'), & \text{every 50 iterations,} \\ (1-\lambda)L_1(p_i, \hat{p}_i) + \lambda L_{\text{D-SSIM}}(p_i, \hat{p}_i) + \beta L_{\text{Edge}}(p_i, \hat{p}_i) + \eta L_{\text{Frequency}}(p_i, \hat{p}_i), & \text{otherwise,} \end{cases} \tag{14}$$

where $\hat{p}_i'$ denotes the pixel in the attention-enhanced image. The $\lambda$, $\beta$ and $\eta$ are the respective weights assigned to the loss components and they are set to 0.2.

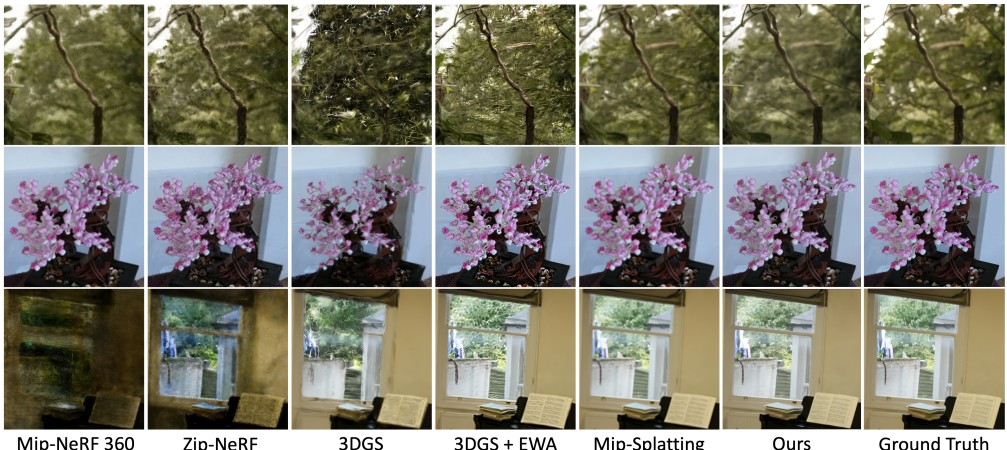

Mip-NeRF 360 · Zip-NeRF · 3DGS · 3DGS + EWA · Mip-Splatting · Ours · Ground Truth

Figure 4: **Qualitative Comparison Results on the Mip-NeRF 360 Dataset (Barron et al., 2022).** These models were trained using images downsampled by a factor of eight and then rendered at full resolution to depict the quality of zooming in and close-ups. In contrast to previous approaches, our model achieves a higher level of accuracy and detail than other models and can render images that are almost identical to the ground truth.

| | SSIM ↑ | | | | | PSNR ↑ | | | | | LPIPS ↓ | | | | |
|---|---|---|---|---|---|---|---|---|---|---|---|---|---|---|---|
| | 1/8 Res. | 1/4 Res. | 1/2 Res. | Full Res. | Avg. | 1/8 Res. | 1/4 Res. | 1/2 Res. | Full Res. | Avg. | 1/8 Res. | 1/4 Res. | 1/2 Res. | Full Res. | Avg. |
| Instant-NGP | 0.748 | 0.645 | 0.620 | 0.690 | 0.676 | 26.85 | 24.90 | 24.15 | 24.40 | 25.08 | 0.238 | 0.373 | 0.452 | 0.466 | 0.382 |
| Mip-NeRF 360 | 0.858 | 0.730 | 0.665 | 0.700 | 0.738 | 29.24 | 25.31 | 24.08 | 24.17 | 25.70 | 0.125 | 0.263 | 0.368 | 0.431 | 0.297 |
| Zip-NeRF | 0.877 | 0.690 | 0.571 | 0.555 | 0.673 | **29.64** | 23.25 | 20.91 | 20.24 | 23.51 | **0.101** | 0.263 | 0.418 | 0.492 | 0.319 |
| 3DGS | 0.882 | 0.735 | 0.616 | 0.622 | 0.714 | 29.25 | 23.44 | 20.80 | 19.52 | 23.25 | 0.105 | 0.242 | 0.396 | 0.483 | 0.307 |
| 3DGS + EWA | 0.882 | 0.773 | 0.673 | 0.646 | 0.744 | 29.34 | 25.87 | 23.69 | 22.83 | 25.43 | 0.112 | 0.235 | 0.371 | 0.448 | 0.292 |
| Ours | **0.883** | **0.811** | **0.749** | **0.766** | **0.802** | 29.35 | **27.22** | **26.41** | **26.25** | **27.31** | 0.111 | **0.210** | **0.301** | **0.389** | **0.253** |

Table 2: **Quantitative Comparison with Baselines on the Mip-NeRF 360 Dataset (Barron et al., 2022).** Each approach is rendered in four different resolutions (1/8, 1/4, 1/2, and the full resolution) after being trained at the lowest resolution (1/8). Our approach produces similar results at the 1/8 resolution and outperforms other models at 1/2, 1/4, and full resolutions.

## 4 EXPERIMENTS

### 4.1 BASELINES

We selected Mip-Splatting (Yu et al., 2024) and 3D-GS (Kerbl et al., 2023) as our primary baseline due to their established state-of-the-art performance in novel view synthesis. In our evaluation, we included several other prominent techniques, such as Mip-NeRF360 (Barron et al., 2022), Mip-NeRF (Barron et al., 2021), Instant-NGP (Müller et al., 2022), Zip-NeRF (Barron et al., 2023), Scaffold-GS (Lu et al., 2024), SuGaR (Guédon & Lepetit, 2024), TensoRF (Chen et al., 2022), and Tri-MipRF (Hu et al., 2023). We also considered NeRF (Mildenhall et al., 2021) and 3DGS + EWA (Zwicker et al., 2001) for further comparison. They are the most representative models.

### 4.2 DATASETS AND METRICS

We carried out an extensive evaluation of multiple scenes sourced from publicly available datasets, including a dataset that features a division of a large scene. Specifically, we assessed our method using seven scenes drawn from Mip-NeRF360 (Barron et al., 2022), the synthetic Blender dataset (Mildenhall et al., 2021), a Villa scene and Mill-19 dataset (Turki et al., 2022). The evaluation metrics we report include Peak Signal-to-Noise Ratio (PSNR), Structural Similarity Index Measure (SSIM) (Wang et al., 2004), and Learned Perceptual Image Patch Similarity (LPIPS) (Zhang et al., 2018). For Mip-NeRF360 and Blender datasets, we present the average values of these metrics across all scenes to provide a comprehensive overview of our approach's performance.

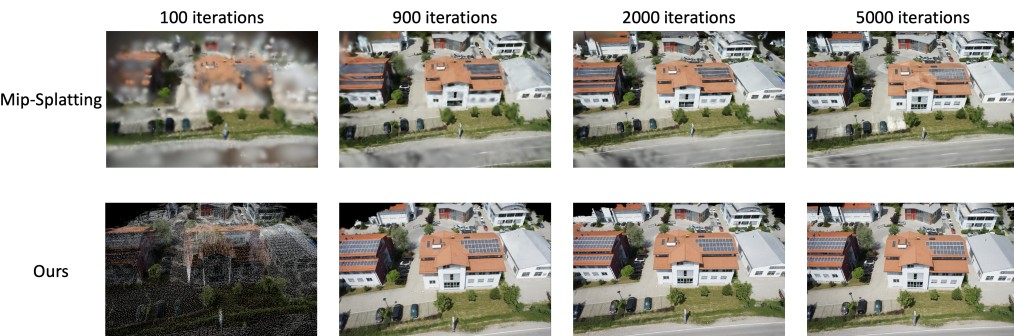

Figure 5: **Training Progression on the Villa Dataset.** We present the quality of the reconstructed villa scene at different training iterations. Compared to the SoTA Mip-Splatting (Yu et al., 2024), our method not only converges faster but also achieves better reconstruction quality with less noise.

| Model | SSIM ↑ | PSNR ↑ | LPIPS ↓ |
|---|---|---|---|
| NeRF | 0.611 | 23.77 | 0.452 |
| Mip-NeRF | 0.621 | 23.99 | 0.439 |
| Mip-NeRF 360 | 0.795 | 27.63 | 0.233 |
| Instant NGP | 0.709 | 25.59 | 0.299 |
| Zip-NeRF | 0.831 | **28.38** | **0.196** |
| 3DGS | **0.832** | 27.69 | 0.217 |
| 3DGS + EWA | 0.818 | 27.74 | 0.214 |
| Scaffold-GS | 0.802 | 27.63 | 0.235 |
| SuGaR (without mesh) | 0.788 | 26.77 | 0.238 |
| Mip-Splatting | 0.827 | 27.79 | 0.203 |
| Ours | 0.825 | 27.69 | 0.208 |

Table 3: **Quantitative Comparison with Baselines on the Mip-NeRF 360 Dataset (Barron et al., 2022).** All approaches are trained and rendered at the same resolution. Our model presents comparable results with other baselines.

## 4.3 RESULT ANALYSIS

**Comparison on the Blender Dataset** Following prior work (Yu et al., 2024), we trained our model on scenes at their original resolution and rendered them at four different resolutions: original, 1/2, 1/4, and 1/8. The quantitative results are detailed in Table. 1 which shows our method outperforms baselines. Our analysis includes NeRF-based (Mildenhall et al., 2021) and 3DGS-based (Kerbl et al., 2023) methods that highlight consistent performance gains across all resolutions, especially at lower resolutions.

**Comparison on the Mip-NeRF 360 Dataset** In our experiments, we trained models on data downsampled by a factor of eight, and then rendered images at different resolutions (1/8, 1/4, 1/2, and original resolution). As illustrated in Table. 2, our method matches prior work at the training resolution (1/8) and significantly outperforms existing state-of-the-art methods at higher resolutions (1/4, 1/2, and original). Fig. 4 demonstrates that our approach renders high-fidelity images without introducing high-frequency artifacts. This is in stark contrast to Mip-NeRF 360 (Barron et al., 2022) and Zip-NeRF (Barron et al., 2023), which tend to falter at higher resolutions due to their MLP architectures' limitations in managing unrepresented frequencies during training. Moreover, the 3DGS method (Kerbl et al., 2023) often yields significant degradation artifacts due to its reliance on dilation processes. Although the 3DGS + EWA method (Zwicker et al., 2001) mitigates some issues, it still produces noticeable high-frequency artifacts. Our method avoids these issues and more accurately represents the ground truth. Additionally, our method effectively reduces blurred artifacts in Mip-splatting (Yu et al., 2024). We further tested our method using the Mip-NeRF 360 dataset, following the protocol where models are trained and evaluated at the same scale. We downsampled indoor scenes by a factor of two and outdoor scenes by a factor of four. The results are detailed in Table. 3, which show that our method achieves performance comparable to both 3DGS (Kerbl et al., 2023) and 3DGS + EWA (Zwicker et al., 2001). This confirms our method's consistent performance across a range of different conditions.

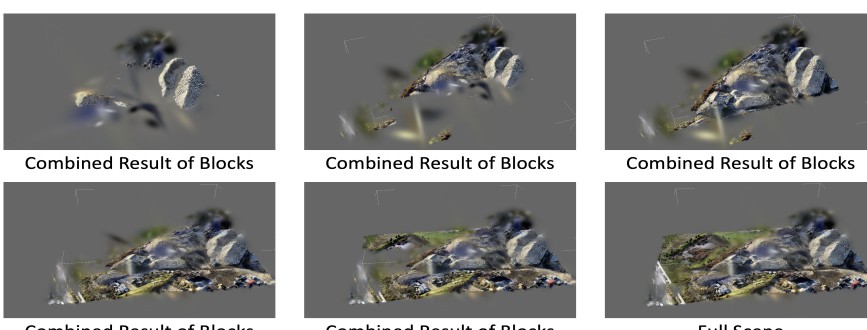

Figure 6: **Reconstructed Result on the Large Scene Dataset (Mill-19)** (Turki et al., 2022). We divide the large scene into individual blocks for separate reconstruction. Here, we display the re-combined results of multiple blocks and the result of the full scene.

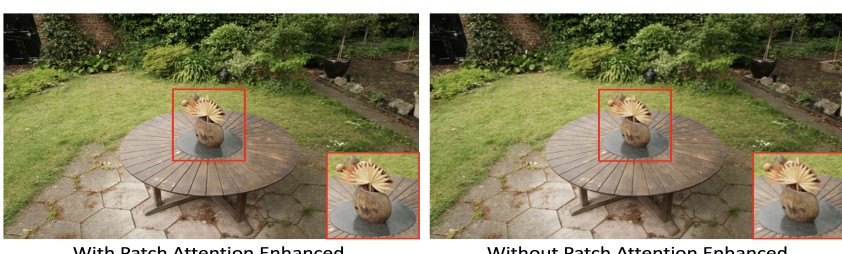

Figure 7: **Ablation of Gaussian Patch Attention Strategy.** We present an ablation study of our model trained on the Garden scene (Barron et al., 2022), comparing results at 30k iterations with and without the application of the Gaussian Patch Attention Enhancement Strategy.

**Comparison on the Villa Dataset**    In the Villa Dataset experiment, we evaluated the training progression of our model against Mip-Splatting (Yu et al., 2024), with both models trained at the original resolution. We presented the results in Fig. 5, showing the performance of both models at various training stages: 100, 900, 2000, and 5000 iterations. Our model showed significant improvements by the 900th iteration. At the same stage, scenes produced by Mip-Splatting (Yu et al., 2024) were still blurry and of lower quality. This difference in performance can be attributed to our Gaussian Constraints Strategy, which effectively controls the growth of Gaussian spheres, leading to faster convergence and superior reconstruction quality. Even after 5000 iterations, the finer details like the roof, windows, and exterior walls reconstructed by Mip-Splatting remained significantly less detailed compared to the achievements of our model in just 900 iterations.

**Evaluation on the Large Scene Dataset**   In our study, we addressed the challenges of reconstructing large scenes like small towns or city-scale environments, which are unmanageable for traditional 3DGS-based (Kerbl et al., 2023) and NeRF-based (Mildenhall et al., 2021) models due to memory constraints and long optimization times. We used the Mill-19 Rubble scene (Turki et al., 2022), which had excessively noisy point clouds requiring reprocessing and selective image filtering. We subdivided the scene, which contained over 1,700 images, into 64 blocks. Each block was independently trained with 200 to 500 images. This reduced memory demands and allowed efficient parallel training in just 20 minutes. Our reconstruction results depicted in Fig. 6, show the seamless reassembly of all blocks which preserves the continuity of the large-scale scene. This method contrasts with previous models, which failed to directly reconstruct large scenes and compromised on reconstruction quality by randomly selecting a subset of images for training.

## 4.4 ABLATION STUDY

### 4.4.1 PATCH ATTENTION ENHANCEMENT

We examined the impact of omitting the Patch Attention strategy from our model. As shown in Fig. 7, removing this strategy leads to noticeable degradation in rendering quality, especially in

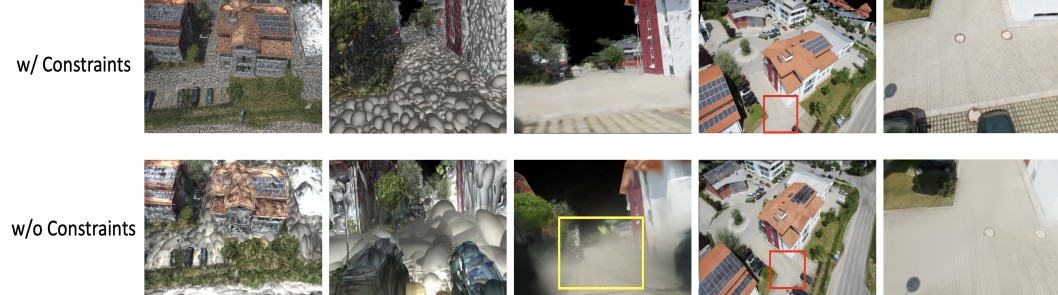

w/ Constraints

w/o Constraints

Figure 8: **Ablation of Gaussian Sphere Constraints Strategy.** We present an ablation study of our model trained on the Villa scene, comparing results at 5k iterations with and without the application of the Gaussian Sphere Constraints Strategy. This strategy reduces the "air walls" problem.

| | Villa | | | Garden | | |
| | SSIM ↑ | PSNR ↑ | LPIPS ↓ | SSIM ↑ | PSNR ↑ | LPIPS ↓ |
|---|---|---|---|---|---|---|
| None | 0.855 | 25.30 | 0.202 | 0.832 | 26.81 | 0.171 |
| w/ Gaussian Constraints | 0.892 | 25.97 | 0.125 | 0.877 | 27.75 | 0.102 |
| w/ Patch Attention | 0.889 | 26.31 | 0.138 | 0.874 | 27.69 | 0.105 |
| Full model | **0.893** | **26.43** | **0.121** | **0.887** | **27.76** | **0.100** |

Table 4: **Ablation Study: Patch Attention Enhancement and Gaussian Sphere Constraints.** We present quantitative results for the Villa and Garden scenes (Barron et al., 2022), trained for 30,000 iterations. Both scenes were downsampled by a factor of four and rendered at the same resolution.

image details. Without Patch Attention, images exhibit blur effects due to high-frequency dilation issues. To quantitatively evaluate this impact, we referred to Table. 4, which compares performance metrics with and without this enhancement. The results clearly indicate improvements across all metrics when the Patch Attention strategy is employed, significantly enhancing the model's ability to produce detailed and sharp renderings by focusing on edge information.

### 4.4.2 GAUSSIAN SPHERE CONSTRAINTS

We assessed the importance of Gaussian Sphere Constraints by removing them from our model. As shown in Fig. 8, models rendered without these constraints exhibit oversized Gaussian spheres, which result in information loss and reduce the overall quality of the renderings. In 3D scenes, these oversized spheres often create "air walls" in detail-heavy areas. Implementing Gaussian Sphere Constraints allows us to effectively control the growth and size of these spheres, enhancing detailed depiction within the scene. The comparative images in Fig. 8, especially in the lower two layers, clearly demonstrate the loss of detail in models rendered without this strategy. These images highlight how the constrained Gaussian spheres maintain finer details, leading to more precise and realistic renderings. Additionally, as indicated in Table. 4, the inclusion of Gaussian Sphere Constraints significantly improves performance metrics.

## 5 CONCLUSION

In this paper, we present GaussianFocus, an enhanced model derived from traditional 3D Gaussian Splatting. It features three key innovations: Patch Attention Enhancement, Gaussian Constraints Strategy and the subdivision of large-scale scenes into manageable blocks for individual training. These innovations aim to refine detail representation, enhance reconstruction quality and reduce the "air walls" problem. The approach of subdividing large scenes into manageable blocks overcomes the limitations inherent in traditional 3DGS-based methods, which struggle with extensive scenes. Experimental results demonstrate that GaussianFocus competes well with state-of-the-art methods at a single scale and excels across multiple scales, providing superior detail accuracy and reconstruction quality.

REPRODUCIBILITY STATEMENT

All the results reported in the paper are reproducible. We submit the code and include all the implementation details in the Abstract and Appendix A.

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

## APPENDIX

## A    RELATED WORK

**Volumetric Rendering methods**    Volumetric approaches utilize structures such as multiplane images, voxel grids or neural network models to depict scenes as continuous functions that define their volume, density, and colour characteristics. The introduction of Neural Radiance Fields (NeRF) (Mildenhall et al., 2021) marked a significant advancement in scene representation technology. This method employs a multilayer perceptron (MLP) to parameterize a continuous volumetric function. This parameterization facilitates the creation of photorealistic images that exhibit precise details. These details and effects are dependent on the viewer's perspective, achieved through volumetric ray tracing. Nevertheless, the application of the vanilla NeRF model is hindered by its high demand for computational power and memory. To overcome these challenges, subsequent research has sought to refine NeRF's efficiency and extend its scalability. Such improvements have been achieved through the implementation of discretized or sparse volumetric frameworks, such as voxel grids and hash tables. These frameworks (Chen et al., 2022; Karnewar et al., 2022; Sun et al., 2022; Müller et al., 2022; Fridovich-Keil et al., 2022) are crucial as they hold learnable features that act as positional encodings for 3D coordinates. Additionally, these methods employ hierarchical sampling techniques (Barron et al., 2022; Reiser et al., 2021; Yu et al., 2021) and utilize low-rank approximations (Chen et al., 2022). Despite these enhancements, the dependence on volumetric ray marching continues, which leads to compatibility challenges with traditional graphics equipment and systems primarily engineered for polygonal rendering. Additionally, recent innovations have adjusted NeRF's approach to geometry and light emission representation, improving the rendering of reflective surfaces (Verbin et al., 2022) and enabling more effective scene relighting by separately addressing material and lighting attributes (Kuang et al., 2022; Srinivasan et al., 2021; Zhang et al., 2021).

**Point-based Rendering methods**    Point-based rendering methods leverage point clouds as fundamental geometric units for the visualization of scenes. The typical methods (Botsch et al., 2005; Sainz & Pajarola, 2004) involve using graphical APIs and GPU-specific modules to rasterize these unstructured point sets at a constant size. Despite the rapid rendering and flexibility in managing changes in topology, this method is prone to forming holes and outliers, which frequently result in rendering artifacts. To address these gaps, research on differentiable point-based rendering has become prevalent, aiming to precisely model the geometry of objects (Insafutdinov & Dosovitskiy, 2018; Gross & Pfister, 2011; Yifan et al., 2019; Lin et al., 2018; Wiles et al., 2020). Research has examined the use of differentiable surface splatting in studies like (Yifan et al., 2019; Wiles et al., 2020), in which points are interpreted as larger-than-one-pixel geometric objects such as surfels, elliptic shapes, or spheres. Methods (Aliev et al., 2020; Kopanas et al., 2021) have enriched point features with neural network capabilities and processed them through 2D CNNs for visualization.

In contrast, Point-NeRF (Xu et al., 2022) has demonstrated superior capabilities in synthesizing new views of high quality using volume rendering, incorporating strategies like region growth and point reduction during its optimization phase. However, this technique is limited by its dependence on volumetric ray-marching, impacting its display speed. Remarkably, the 3DGS (Kerbl et al., 2023) framework employs directionally dependent 3D Gaussians for three-dimensional scene depiction. This method utilizes structure from motion (SfM) (Snavely et al., 2006) to initialize 3D Gaussians and optimizes a 3D Gaussian as a volumetric model. Subsequently, it projects this model onto 2D surfaces to facilitate rasterization. 3D-GS uses an $\alpha$-blender to merge pixel colours effectively. This technique results in high-fidelity outputs with detailed resolution, enabling rendering at real-time speeds.

**Large-scale Scene Reconstruction**    In the past several decades, remarkable advancements have been made in the domain of image-based reconstruction of extensive scenes. Numerous research efforts (Pollefeys et al., 2008; Schonberger & Frahm, 2016; Zhu et al., 2018; Snavely et al., 2006) have leveraged the structure-from-motion (SfM) (Snavely et al., 2006) method to derive camera orientations and generate sparse point clouds. Following these initiatives, additional studies (Furukawa et al., 2010; Goesele et al., 2007) have succeeded in producing dense point clouds or triangular meshes via multi-view stereo (MVS) processes. Concurrently, as Neural Radiance Fields (NeRF) (Mildenhall et al., 2021) gain prominence for generating photorealistic perspectives in contemporary visual synthesis, a plethora of adaptations have surfaced. These aim to increase reconstruction quality (Barron et al., 2021; 2022; 2023; Wang et al., 2021; 2023; Yariv et al., 2021), accelerate rendering (Chen et al., 2022; Fridovich-Keil et al., 2022; Müller et al., 2022; Reiser et al., 2021; Yu et al., 2021), and extend capabilities to dynamic scenarios (Cao & Johnson, 2023; Gao et al., 2022; Weng et al., 2022; Huang et al., 2024). Among these, several methods (Tancik et al., 2022; Turki et al., 2022; Xu et al., 2023; Zhenxing & Xu, 2022) have scaled NeRF to accommodate expansive scenes. Specifically, Block-NeRF (Tancik et al., 2022) segments urban landscapes into several blocks, assigning view-specific training based on geographic location. Alternatively, Mega-NeRF (Turki et al., 2022) introduces a grid-oriented partitioning technique, linking each image pixel to various grids intersected by its corresponding ray. Different from heuristic partitioning methods, Switch-NeRF (Zhenxing & Xu, 2022) has pioneered a mixture-of-experts NeRF framework to master scene segmentation. Conversely, Grid-NeRF (Xu et al., 2023) synergizes NeRF-based and grid-based strategies without segmenting the scene. Despite these improvements significantly elevating rendering precision over conventional methods, they often render slowly and lack finer details. In a recent development, 3D Gaussian Splatting (3DGS) (Kerbl et al., 2023) has been introduced. It provides an explicit, high-definition 3D representation that supports real-time rendering. However, these traditional 3DGS methods (Kerbl et al., 2023; Yu et al., 2024; Lu et al., 2024; Guédon & Lepetit, 2024) have been shown to consume significant resources when applied to extensive scenes, such as urban environments or scenic landscapes. This is primarily due to the considerable memory and graphics memory demands necessary for initial scene processing and the creation of Gaussian spheres. Previous methodologies (Kerbl et al., 2023; Yu et al., 2024; Lu et al., 2024) for reconstructing large scenes typically relied on selecting a subset of images for training and then regenerating point clouds and viewpoints using COLMAP (Schonberger & Frahm, 2016), which employs Structure-from-Motion (SfM) and Multi-View Stereo (MVS) techniques to derive camera positions and 3D structures from images. However, this approach proved to be inherently non-generalizable. The primary issue was the lack of effective scene segmentation, which led to random retention of images. Consequently, this resulted in fragmented reconstruction outcomes. Moreover, these approaches lead to disparate Gaussian outcomes, which could not be merged effectively. Each batch of partial images remained with isolated training results that lacked collective significance. Additionally, the use of incomplete image sets in training often resulted in inadequate COLMAP (Schonberger & Frahm, 2016) results due to the failure to accurately select all required viewpoints for a comprehensive scene reconstruction. Our GaussianFocus successfully overcomes the limitations of the 3DGS-based methods (Kerbl et al., 2023; Yu et al., 2024; Lu et al., 2024) in training large-scale scenes through the introduction of innovative designs that efficiently subdivide, optimize, and integrate these scenes.

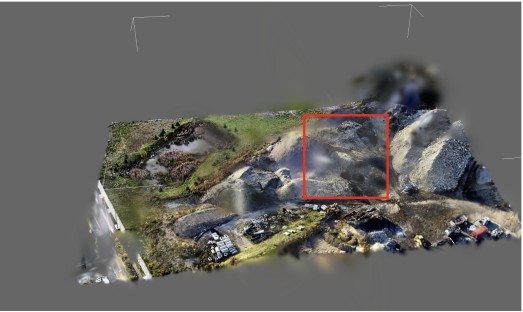

Figure 9: **Limitation:** The result of the recombined scene will contain the boundary artifacts of the result of the previous small block.

## B    IMPLEMENTATION

Our approach is developed on the foundation of the open-source 3DGS code (Kerbl et al., 2023). Adhering to the protocol established in (Kerbl et al., 2023), we train our models and baselines for 30,000 iterations over all scenes, utilizing the same Gaussian density control strategy, training pipelines and hyperparameters. Furthermore, patch attention is utilized to enhance reconstruction quality every 50 iterations. We also inspect and constrain the scale matrix $S$ of Gaussian spheres every 1,000 iterations, up to the first 10,000 iterations. We set the kernel size as 0.05 and the loss weight $\lambda = 0.2$.

## C    LIMITATION

Our model integrates the Patch Attention Enhancement feature, which substantially improves the quality of rendered images by meticulously calculating attention values. While this method enhances detail recognition and overall image fidelity, it also significantly increases the memory demands of the model. This elevated memory consumption has the potential to trigger out-of-memory errors during the training phase, particularly with complex or large-scale scenes. To address this limitation, future versions of the model could explore alternative computational methods or more efficient data structures, which might reduce the memory requirements while maintaining or even enhancing the model's performance. Another challenge arises in the reconstruction of large scenes where the final assembly of individual blocks can lead to complications. Specifically, the boundaries of each block may overlap, causing visible disruptions in the continuity of the scene. These overlaps often manifest as clusters of disorganized Gaussian spheres at the edges, which are evident in the reconstructed images shown in Fig. 9. This not only affects the aesthetic quality of the renders but also detracts from the model's utility in practical applications. In the future, it may be beneficial to design an algorithm that removes Gaussian spheres at the boundaries of each block. This would enhance the quality of the final assembled large scene and ensure a more natural and seamless appearance. Currently, our model is implemented using the Pytorch framework. While Pytorch provides a robust platform for developing deep learning models, it may not offer the most efficient management of large-scale data and complex computations involved in our model. Transitioning our model to a CUDA-based implementation could significantly improve efficiency.

