# OpenReview forum: "GaussianFocus: Constrained Attention Focus for 3D Gaussian Splatting"
_ICLR.cc/2025/Conference — ICLR 2025 Conference Withdrawn Submission_

### Official Review · Reviewer_zbn1 · 2024-10-26

**Soundness:** 1
**Presentation:** 1
**Contribution:** 1
**Rating:** 1
**Confidence:** 5

**Summary:**

In response to the artifacts in large-scale scene reconstruction, GaussianFocus introduces a patch attention algorithm to enhance edges and high-frequency areas. Additionally, the authors propose a partition strategy to accelerate the training process. The results appear promising.

**Strengths:**

1. The paper is easy to understand and follow.
2. GaussianFocus introduce an impressive patch attention module to further improve the rendering quality of Gaussian splatting.

**Weaknesses:**

1. **Lack of novelty.** Aside from the 3DGS-based patch attention algorithm, I struggle to find any significant innovation in the paper. Moreover, I question the validity of this algorithm due to the insufficient ablation study. Constraints on the shapes of GS spheres have been widely adopted, such as volume regularization in Scaffold-GS, covariance regularization in 3DGS-MCMC, and scale adjustment in Hierarchical-3DGS. The idea of subdividing and then merging to deal with large-scale scene is similar with Vast-Gaussian and Hierarchical-3DGS.
2. **Strange typography.** Placing the related work in the appendix is unconventional. Furthermore, while the authors include several irrelevant papers in the related work section, they fail to cite the most relevant ones, such as Vast-Gaussian, DoGaussian, and Hierarchical-3DGS.
3. **Grammatical error in writing.**
    1. In line 42, the sentence is better to be written as “This technique refines a set of 3D Gaussians, initialized using Structure from Motion (SfM) (Snavely et al., 2006), to model scenes with inherent volumetric continuity. This facilitates fast rasterization by projecting them onto 2D planes.”
4. **Logical Error.**
    1. In line 47, to address the artifacts issue, Mip-splatting utilizes both 3D smoothing and 2D Mip filters. Scaffold-GS introduces regularly spaced anchors as a structural prior, which improves the arrangement of Gaussian primitives, not just “utilize anchor points to initialize 3D Gaussians”. The authors need to delve deeper and identify the key reasons for the improvement.
5. **Experiments Error**
    1. **Baseline selection.** The paper focuses on rendering improvements, yet it uses several NeRF methods (whose rendering performance is already well-known) and SuGar (which is designed for surface reconstruction) as baselines. This choice seems neither meaningful nor representative overall. Instead, the authors should compare their approach with methods like Octree-GS and 3DGS-MCMC, which are among the best for rendering.
    2. **Dataset Selection.** What is the villa scene? Did the authors collect it themselves? Although the authors claim that the method is specifically designed for large-scale scenes, they only compare it using a villa scene and the Mill-19 dataset, which seems unconvincing.
    3. **Weird Quantitative Results.** GaussianFocus is not designed for anti-aliasing, yet the authors train and render at four different resolutions, which seems quite odd. Additionally, in Table 1 and Table 2, the authors do not compare their method with Mip-Splatting or its derivatives. In Table 3, the performance is not state-of-the-art, and the results for Scaffold-GS appear too low—it is unreasonable for its performance to be worse than 3DGS.
    4. **Weird implementation.** In line 475, the authors state, 'We subdivided the scene, which contained over 1,700 images, into 64 blocks. Each block was independently trained with 200 to 500 images.' However, no method typically subdivides 1,700 images into more than 9 blocks. Using so many chunks creates excessive and wasteful redundancy.
    5. **Insufficient ablation study.** The authors conduct the ablation study on only two scenes in Table 4, and in Figure 8, they display qualitative results at 5K iterations instead of 30K iterations.
6.  **No demos submitted**

**Questions:**

This paper is not up to iclr standards at all. The authors have to particularly polish the exp part.

---

### Official Review · Reviewer_6nEe · 2024-10-31

**Soundness:** 1
**Presentation:** 2
**Contribution:** 1
**Rating:** 3
**Confidence:** 5

**Summary:**

This paper introduces a 3D Gaussian Splatting (3DGS)-based method designed to enhance the rendering quality of standard 3DGS. It proposes a patch-based algorithm to improve rendering quality and a Gaussian constraint strategy to reduce redundancy. For large-scale scenes, the method employs a subdivision strategy to split the scene into smaller, mergeable blocks for separate 3DGS training.

However, the patch attention strategy lacks clear motivation and theoretical justification, and the provided ablation study appears to weaken its perceived effectiveness. For the Gaussian constraint strategy, the approach uses a basic manual adjustment by applying a scalar to reduce the size of Gaussian spheres exceeding a set threshold.

Additionally, the comparisons with baseline methods are somewhat confusing and require further clarification to be fully understood.

**Strengths:**

1. The paper is well-structured and presents its ideas clearly.

2. Extensive experiments are conducted on multiple datasets.

3. The straightforward Gaussian constraint approach demonstrates its effectiveness.

**Weaknesses:**

1. **Patch Attention Strategy Issues:** The patch attention strategy lacks a strong motivation and theoretical grounding, and the ablation study provided seems to diminish its perceived effectiveness. In Equation 7, the rendered image is directly multiplied by the attention map to produce an “attention-enhanced” image. However, this operation appears flawed; in neural rendering, the objective is to minimize the difference between the rendered and ground-truth images. By multiplying the rendered image by an additional attention map, the resulting image loses physical meaning, potentially causing optimization inconsistencies. Furthermore, in Table 4, results with patch attention are poorer than those with Gaussian constraints, calling the effectiveness of this module into question.

2. **Lack of Novelty in Gaussian Constraint Strategy:** The Gaussian constraint strategy lacks innovation, as it is a simple manual scaling operation that reduces large Gaussian spheres by applying a fixed scalar.

3. **Inconsistencies in Baseline Comparisons:** Baseline comparisons are presented on the Mip-NeRF 360 dataset in both Table 2 and Table 3. However, the results for the proposed method and the baselines appear inconsistent between these tables. The authors should clarify the reason for these discrepancies.

4. **Effectiveness Concerns:** In Table 4, the proposed method shows poorer results compared to 3DGS, raising concerns about the overall effectiveness of this approach.

**Questions:**

1. What are the definitions of G_x() and G_y() in Eq.13 ?
2. Why adopt \alpha=0.2 and \tau=0.3 in Eq.8 ?

---

### Official Review · Reviewer_b4Hw · 2024-11-02

**Soundness:** 3
**Presentation:** 2
**Contribution:** 2
**Rating:** 3
**Confidence:** 5

**Summary:**

The paper applies the attention-enhanced rendered image and the edge and frequency loss to improve the 3DGS optimization. Meanwhile, the authors impose constraints on Gaussian spheres to mitigate the occurrence of “air walls” problems in large-scale scene reconstruction. In the experiment section, the authors show that the proposed method surpasses previous 3DGS reconstruction, and demonstrate the effectiveness of the proposed two components through ablation studies.

**Strengths:**

- The proposed two components are clear to me. The ablation studies on the two scenes have shown the effectiveness of the proposed components.

- The attention-enhanced rendered image is new to me.

**Weaknesses:**

- The authors also claim scene partition as one of the contributions of the manuscript. However, this part is two subtle and still has limitations. I would suggest removing this part from the major contribution of the paper.

- An ablation study is missing for the edge and frequency losses. How does each loss contribute to the final performance gain?

- The ablation study is crucial for a paper with incremental enhancements. However, the authors only conducted the ablation study on two scenes that are uncommonly used, making it suspicious about the effectiveness of the proposed component. I suggest the authors conduct the ablation study using 1-2 full datasets, e.g., NeRF-Synthetic, Mip-NeRF 360, DTU, or Tanks and Temples.

- Table 2 and Table 3: why there are two different quantitative evaluations on Mip-NeRF 360 datasets?

- The visualization of the figures could be further improved. For example, the differences in Fig. 1, Fig. 6, and Fig. 7 are unobvious and could be further highlighted; Fig. 8 is vertically distorted.

**Questions:**

Is the attention-enhanced rendered image applicable to general differentiable rendering? It would be more persuasive if the method could be also applied to NeRF optimization.

---

### Official Review · Reviewer_ZbEz · 2024-11-03

**Soundness:** 2
**Presentation:** 2
**Contribution:** 2
**Rating:** 3
**Confidence:** 5

**Summary:**

This paper introduces a novel approach that integrates a Patch Attention algorithm, Gaussian constraints and Block Partition within the 3D Gaussian Splatting (3DGS) framework, aimed at improving rendering quality and enabling large-scale scene reconstruction. Experiments shows the proposed method converges faster and achieves higher rendering quality.

**Strengths:**

+: The enhancement in high frequency area makes sense to me, which is also proved by the quantitative comparison in tables and figures.

+: Every details of the method are well-presented, I can understand them easily.

**Weaknesses:**

-: The paper proposes three contributions: Patch Attention, Scale Constraint, Block Partition and put the first one in the title. However,
 (1) According to the Table 4, it seems that the Scale Constraint is the most simple and effective part? And actually this is somewhat wide-known strategies. Many GS works restrict the scale to be small with a scale loss. Within 3DGS, there is a trick to eliminate gaussians larger than a certain scale. And most similarly, in Scaffold-GS, scales large than 0.05 would be reset to 0.05.
 (2) the effectiveness of the Patch Attention is not well-verified. According to Table 4, although the PSNR from using Patch Attention is high, the constraint on scale seems contributes more to the image structure. This potentially implies that the Patch Attention module benefits from causing more gaussian point by providing a large loss value -> large gradient -> trigger more densification. If this speculation establishes, then a simpler way to enhance the quality would be slightly increasing the weight of \beta and \eta in Eq 14, without using the computation-intensive Patch Attention.

-: The three contributions appear quite fragmented and don’t come together as a single, clear main contribution. Author should polish and re-organize the paper. (This is actually just a suggestion, not relevant to the rate.)

-: Lacking overhead analysis after adding these designs.

**Questions:**

Please refer to the weakness.

---

### Note · Authors · 2024-11-13

I have read and agree with the venue's withdrawal policy on behalf of myself and my co-authors.